# Optimal Sensor Placement in Hydraulic Conduit Networks: A State-Space Approach

**Caspar V. C. Geelen** [1,2], **Doekle R. Yntema** [2] , **Jaap Molenaar** [1] **and Karel J. Keesman** [1,2,*]

1 Department for Mathematical and Statistical Methods—Biometris, Wageningen University, P.O. Box 16, 6700 AA Wageningen, The Netherlands; caspar.geelen@wur.nl (C.V.C.G.); jaap.molenaar@wur.nl (J.M.)
2 Wetsus, European Centre of Excellence for Sustainable Water Technology, Oostergoweg 9, P.O. Box 1113, 8900 CC Leeuwarden, The Netherlands; Doekle.yntema@wetsus.nl
* Correspondence: karel.keesman@wur.nl; Tel.: +31-317-48-3780

**Abstract:** Conduit bursts or leakages present an ongoing problem for hydraulic fluid transport grids, such as oil or water conduit networks. Better monitoring allows for easier identification of burst sites and faster response strategies but heavily relies on sufficient insight in the network's dynamics, obtained from real-time flow and pressure sensor data. This paper presents a linearized state-space model of hydraulic networks suited for optimal sensor placement. Observability Gramians are used to identify the optimal sensor configuration by maximizing the output energy of network states. This approach does not rely on model simulation of hydraulic burst scenarios or on burst sensitivity matrices, but, instead, it determines optimal sensor placement solely from the model structure, taking into account the pressure dynamics and hydraulics of the network. For a good understanding of the method, it is illustrated by two small water distribution networks. The results show that the best sensor locations for these networks can be accurately determined and explained. A third example is added to demonstrate our method to a more realistic case.

**Keywords:** optimal sensor placement; state-space representation; observability gramian; water distribution network

## 1. Introduction

Hydraulic models are an essential tool for ensuring safe, reliable, affordable, and continuous delivery of fluids, such as water or oil, to end-users [1]. Due to the physical size and complexity of most conduit networks, the actual operational conditions of these grids are hard to monitor [2]. Besides serving as a digital twin of the network, model simulations of the real system can be used to predict and forecast, in real-time, network flows and pressures under varying hydraulic scenarios, valve configurations, or conduit leakages [3–6]. In addition, distribution system modelling can help optimize network design, sensor and actuator placement, or facilitate operation of the network through testing of control strategies of pumping and valve configurations [2–4,6–11]. Models, therefore, act as an active tool for network management and leakage control, instead of the classical passive approach of solely reacting when a defunct asset is detected.

In order to successfully deploy these models, accurate, and up-to-date, insight into the network is required in the form of real-time measurements from sensors placed throughout the network. Sensor placement, operation, and maintenance is costly, meaning there is a tradeoff between network information gain and sensor costs. A wireless sensor network of as few as possible flow and pressure sensors, at key positions within the network, is of vital importance for optimal network management. Therefore, optimal sensor placement poses an ongoing challenge in hydraulic conduit networks.

Various studies have been conducted with the goal of maximizing the diagnostic performance of a system under budgetary constraints by means of applying optimal sensor placement. Current methods often consider optimal sensor placement with regards

to leak detectability by simulating hydraulic scenarios with leakages [2,11–14]. These studies are mostly based on simulating leakages at various locations within a hydraulic network. The change in each state (pressure/flow), as a consequence of these leakages, is captured in a binarized sensitivity matrix, represented by, e.g., a Jacobian or forward finite differences matrix. The potential sensor locations are then ranked based on the number of burst locations for which significant sensitivities are found [15,16]. This approach can be expanded in several directions, for example, by using artificial bursts achieved by opening fire hydrants in various areas of the network instead of model simulated bursts [9], by considering demand uncertainty [7], by not binarizing the sensitivity matrix [10,17], by using optimization techniques instead of iterative techniques to determine the optimal sensor location [14], or in combination with a leakage localization algorithm [11]. In addition to these studies on hydraulic networks, observability and sensor placement studies on electrical networks have also been performed [18–20].

Although practical and efficient, the performance of existing sensor placement techniques, based on virtual leakage simulation, is highly dependent on the accuracy of the hydraulic model. The estimated sensor placement is very sensitive to uncertainty in demand estimates, model parameters, measurement noise, and asset properties, and has to consider different leakage locations and fluid loss rates, in order to provide accurate sensor placement suggestions [21]. Considering all these factors does, not only, result in a high dimensional problem and extensive simulations of hydraulic scenarios but also in high cumulative uncertainties, which exponentially worsens when considering simultaneous placement of multiple sensors. Recent publications, on the topic of tackling this high dimensional and highly uncertain problem, all suggest to invest more research into these uncertain factors and focus on development of smart optimization strategies to reduce the computational load [6,7,11,17,22]. Additionally, an extra source of uncertainty in existing theories is that pressure changes are assumed to take place instantaneously, which, especially, for larger networks is too rough an assumption [11,23].

The objective of this study is to investigate the observability of conduit networks and optimal sensor placement designs, only considering the structure of the hydraulic network model, and without a dependence on dynamic network simulations. By not relying on simulation of hydraulic scenarios, no computationally expensive high dimensional optimization is required. In this study, a linearized hydraulic network model is presented in state-space form, with, as model states, the pressures in all network junctions and the flows through all network pipes. Model outputs are the model states corresponding to network junctions and/or conduits where a pressure or flow sensor is installed. Based on a likely hydraulic scenario, with corresponding stationary network flows calculated with an EPANET model, the original non-linear hydraulic network model was linearized. This linearization step enables conventional observability analysis [24,25] and optimal sensor placement based on maximizing the output energy of observability Gramians [26]. Current research often focuses on smart optimization to reduce the computational load associated with simultaneous placement of multiple sensors [6,7,11,17,22]. This study, however, aims to present an alternative sensor placement framework that starts with state-space modelling. The advantages of using a state-space model, for sensor placement in hydraulic conduit networks, are demonstrated by two illustrative examples and one more realistic example with corresponding EPANET models [27].

We also show how the suitability of each conduit and junction, in the model for sensor placement, can be mapped on a graph of the network. This visualization allows for easy identification of optimal regions in the network for sensor placement. This visual information can be used to combine observability function-based optimal sensor placement with other network-specific knowledge regarding sensor placement. The state-space methodology in this paper has been developed using open source software and is equipped with the capacity to transform EPANET model files into state-space models using a Python 3 algorithm.

## 2. Materials and Methods

In order to perform optimal sensor placement in a hydraulic network, based on systems theory, in this study, the network characteristics and dynamics are presented in state-space form. Given the network characteristics, such as conduit lengths, diameters, and roughness, flows and pressures within the hydraulic system can be modelled using continuity and momentum equations for unsteady, nonuniform flow of a slightly compressible fluid in slightly elastic conduits. For each conduit, these assumptions lead to the following set of hyperbolic partial differential equations [1,28]:

$$\frac{\partial}{\partial t}\begin{pmatrix} p \\ V \end{pmatrix} + \begin{bmatrix} V & \rho c^2 \\ \rho^{-1} & V \end{bmatrix} \frac{\partial}{\partial x}\begin{pmatrix} p \\ V \end{pmatrix} = \begin{pmatrix} 0 \\ -g\,sin(\theta) - \frac{fV|V|}{2D} \end{pmatrix} \tag{1}$$

Here, $p$ is the pressure in Pa, $V$ is the flow velocity in $\frac{m}{s}$, $\rho$ is the mass density of the transported fluid in $\frac{kg}{m^3}$, $c$ is the elastic wave velocity in $\frac{m}{s}$, $g$ is the acceleration due to gravity in $\frac{m}{s^2}$, $\theta$ is the angle the conduit makes with the horizontal, with the angle taken positive if the conduits slopes upwards in the flow direction, $f$ is the Darcy–Weisbach friction factor (dimensionless), and $D$ is the diameter of the inside of the conduit in m. The distinction between the magnitude of flow velocity $|V|$ and directional flow velocity $V$ is made to allow for flow in both directions through a conduit. For a thorough observability analysis of systems described by hyperbolic partial differential equations, we refer to [29].

The slope term $g\,sin(\theta)$ is relatively small for most applications and may be neglected. Even if the slope is taken into account, the term will be interpreted as a disturbance and will therefore not influence optimal sensor placement, which solely relies on flow and pressure dynamics, as well as sensor configurations.

Also, in many applications, the convective acceleration terms $V(\partial p/\partial x)$ and $V(\partial V/\partial x)$ are small compared to the other terms and may, therefore, be neglected [1]. However, in this study, we assume slightly compressible fluid in slightly elastic conduits and thus changes in pressure and flowrate with distance $\frac{\partial}{\partial x}\begin{pmatrix} p \\ V \end{pmatrix}$ are not zero. Furthermore, with $p = \rho g(H + z_0)$ and $V = \frac{Q}{A}$, Equation (1) can be expressed in terms of the piezometric head $H = \frac{p}{\rho g} - z_0$ above a specified level $z_0$, and volumetric flow rate $Q$ with conduit's cross-sectional area $A = \frac{1}{4}\pi D^2$ [1,30]. In this state transformation both $\rho$ and $A$ are assumed to be constant. The variation of $\rho$ and $A$ is still indirectly taken into account by using a finite elastic wave velocity $c$. The elastic wave velocity $c$ is a function of various properties of the transported fluid as well as the conduit, but is assumed constant within a pipe, since the changes within a single conduit are assumed small [31]. For water transport without air bubbles through PVC conduits, the wave velocity is estimated as $c = 1200\frac{m}{s}$ [1].

Equation (1) makes use of the empirical Darcy–Weisbach equation to describe friction losses as $\frac{f|V|}{2D}V$, where the Darcy–Weisbach friction factor $f$ is a function of the Reynold's number. When solely considering water transport and assuming constant temperature and viscosity, such as is the case in water distribution networks, the friction losses are not dependent on the Reynold's number according to the empirical Hazen–Williams equation [1]. Expressed in volumetric flow rate and piezometric head, the Darcy–Weisbach friction related flow loss $\frac{8}{\pi^2}\frac{f|Q|}{D^3}Q$ is replaced by the Hazen–Williams friction related flow loss $\frac{\pi}{4}\frac{10.67g|Q|^{0.852}}{C^{1.852}D^{2.8704}}Q$, where $C$ is the conduit-specific dimensionless Hazen–Williams roughness coefficient. Implementing all these assumptions, we can rewrite Equation (1) in the form [1]:

$$\frac{\partial}{\partial t}\begin{pmatrix} H \\ Q \end{pmatrix} + \begin{bmatrix} 0 & \frac{4}{\pi}\frac{c^2}{gD^2} \\ \frac{\pi}{4}gD^2 & 0 \end{bmatrix} \frac{\partial}{\partial x}\begin{pmatrix} H \\ Q \end{pmatrix} = \begin{pmatrix} 0 \\ -\frac{\pi}{4}\frac{10.67g|Q|^{0.852}}{C^{1.852}D^{2.8704}}Q \end{pmatrix} \tag{2}$$

We will apply this system of hyperbolic partial differential equation to a conduit network with $n_i$ junctions and $n_{ij}$ conduits, connecting junctions $i$ and $j$. Assuming $\frac{\partial}{\partial x}\begin{pmatrix} H_{ij} \\ Q_{ij} \end{pmatrix}$ along conduit $ij$ may be approximated by $\begin{pmatrix} \Delta H_{ij}/L \\ \Delta Qij/L \end{pmatrix}$, where $L$ is the length of the conduit in the flow direction, we arrive for the head in junction $i = 1, 2, \ldots, n_i$ at the following equation [32]:

$$\frac{dH_i}{dt} = \sum_{j=1}^{\deg(i)} \left( \frac{4}{\pi} \frac{c^2}{g D_{ij}^2 L_{ij}} \left( Q_{ij(i)} - Q_{ij(j)} \right) \right) \tag{3}$$

The difference between the flow in a conduit $ij$ at conduit start $i$ and end $j$ is a consequence of the slight compressibility of the fluid and the slight elasticity of the conduit. For fluid transport, the difference between flow at beginning and end of a conduit is usually very small. Commonly, large pressure changes, as a result of high elastic wave velocity $c$, are assumed immediate. Hence, both the low flow variation within a conduit and the rapid pressure changes motivate the assumption that pressure changes are instantaneous, implying that the head in each junction is always in steady state, thus $\frac{dH_i}{dt} = 0$ and thus $Q_{ij(i)} = Q_{ij(j)} \equiv Q_{ij}$ [1]. Consequently, under these assumptions, the dynamics of the system would be solely governed by the momentum Equation (2):

$$\frac{dQ_{ij}}{dt} = \frac{\pi}{4} \frac{g D_{ij}^2}{L_{ij}} \left( H_i - H_j \right) - \frac{\pi}{4} \frac{10.67 g |Q|_{ij}^{0.852}}{C_{ij}^{1.852} D_{ij}^{2.8704}} Q_{ij} \tag{4}$$

Although this equation is very suitable for calculation of hydraulic scenarios and thus, for performing optimal sensor placement through the use of burst simulations, significant and measurable pressure transients do occur [31]. Since modern sensors can operate under sampling frequencies higher than once per second, the damping oscillations, as a consequence of pressure transients and friction in the conduits, can be detected. Since large pressure transients, also referred to as water hammers, can cause conduit wear and bursts, it is important to be able to identify where, how often, and to what extent these transients occur in order to identify their causes and adopt a mitigation strategy. For optimal sensor placement, based on observability analysis, we take into account these transients by assuming $\frac{dH_i}{dt} \neq 0$, and assuming a linear relationship between change in flow rate and flow rate throughout each conduit in the flow direction $i \to j$:

$$Q_{ij(i)} - Q_{ij(j)} = \varepsilon Q_{ij} L \tag{5}$$

The relative flow gradient $\varepsilon$ in m$^{-1}$ is small, since the compressibility of fluids and elasticity of conduits are very small in most hydraulic conduit networks. As $\varepsilon$ is unknown, in this study, we assume it is unknown-but-bounded. Thus, $\varepsilon$ is defined on an interval that represents the uncertainty in the values of the compressibility and elasticity. For details about the relative flow gradient, see Appendix A. This assumption (5) yields a system consisting of a linear continuity and a non-linear momentum equation:

$$\begin{aligned} \frac{dH_i}{dt} &= \sum_{j=1}^{\deg(i)} \left( \frac{4}{\pi} \frac{c^2 \varepsilon}{g D_{ij}^2} Q_{ij} \right) \\ \frac{dQ_{ij}}{dt} &= \frac{\pi}{4} \frac{g D_{ij}^2}{L_{ij}} \left( H_i - H_j \right) - \frac{\pi}{4} \frac{10.67 g |Q|_{ij}^{0.852}}{C_{ij}^{1.852} D_{ij}^{2.8704}} Q_{ij} \end{aligned} \tag{6}$$

Notice that Equation (6) is nonlinear with regards to volumetric flow $Q_{ij}$. However, for small perturbations from a specific hydraulic scenario, as a result of steady state computations in EPANET with steady state pressures $\overline{H}$ and flows $\overline{Q}$, the model can be linearized around that hydraulic scenario. Thus, we assume $|Q_{ij}|_{ij}^{0.852} Q_{ij} \approx |\overline{Q}_{ij}|_{ij}^{0.852} Q_{ij}$.

Consequently, the dynamics of the system are then expressed in terms of the 'resistance' $\mathcal{X}_{ij} = \frac{4}{\pi} \frac{c^2 \varepsilon}{g D_{ij}^2}$, 'conductance' $\mathcal{Y}_{ij} = \frac{\pi}{4} g \frac{D_{ij}^2}{L_{ij}}$, and 'friction' $\mathcal{Z}_{ij} = -\frac{\pi}{4} \frac{10.67 g |\overline{Q}|_{ij}^{0.852}}{C_{ij}^{1.852} D_{ij}^{2.852}}$ constants:

$$
\begin{aligned}
\frac{dH_i}{dt} &= \sum_{j=1}^{\deg(i)} \left( \mathcal{X}_{ij} Q_{ij} \right) \\
\frac{dQ_{ij}}{dt} &= \mathcal{Y}_{ij} \left( H_i - H_j \right) + \mathcal{Z}_{ij} \, Q_{ij}
\end{aligned}
\tag{7}
$$

Simulation of a steady-state hydraulic scenario is, thus, required in order to obtain estimates for the linearization points $\overline{Q}_{ij}$. Although EPANET hydraulic simulations do not include pressure dynamics, these dynamics are still taken into account in the state-space model Equation (7).

Notice that Equation (3) and thus also Equation (7) describe the pressure change as a result of gradients in the flow rates. In steady state, for incompressible fluid and non-elastic networks, the continuity equation at each node is given by: $\frac{dH_i}{dt} = \sum_{j=1}^{\deg(i)} \left( Q_{ij} \right) / A_i = 0$. Small deviations in the flow rates through a node, as a result of changing boundary conditions, may lead to small increases or decreases of the pressure in the node, which are also covered by Equation (5). Consequently, as a result of our approximations, for large changes in the hydraulic scenario, new steady states need to be calculated, using, e.g., the EPANET model (Rossman, 2000).

To put these equations in matrix-vector form, we introduce the state vector $x := \left[ H_1, H_2, \ldots, H_{n_i}, Q_1, Q_2, \ldots, Q_{n_{ij}} \right]^T \in \mathbb{R}^n$ with $n = n_i + n_{ij}$, where the first $n_i$ elements contain the heads $H_i$, and the remaining elements the flows $Q_{ij}$. We further introduce the output vector $y = \left[ y_1, y_2, \ldots, y_p \right]^T \in \mathbb{R}^p$ with $p = p_i + p_{ij}$, where the first $p_i$ elements contain the heads $H_i$ of those junctions equipped with a pressure sensor and the remaining elements contain the flows $Q_{ij}$ of those conduits equipped with a flow sensor. In what follows, we assume a pressure sensor is always placed in a junction and a flow sensor halfway on a conduit. Equation (7) can thus be represented in the following form:

$$
\begin{aligned}
\frac{d}{dt} x(t) &= \boldsymbol{A} x(t) + \boldsymbol{B} u(t) \\
y(t) &= \boldsymbol{C} x(t) + \boldsymbol{D} u(t)
\end{aligned}
\tag{8}
$$

where $u(t)$ contains the boundary conditions and in what follows $\boldsymbol{D} = \boldsymbol{0}$. The dynamics of the system are determined by the system specific parameters $\mathcal{X}_{ij}$, $\mathcal{Y}_{ij}$, and $\mathcal{Z}_{ij}$, which make up the elements of the $n \times n$ matrix $\boldsymbol{A}$. The positions of the sensors are specified via the $p \times n$ matrix $\boldsymbol{C}$. For applications of optimal sensor placement, only the system dynamics (matrix $\boldsymbol{A}$) and the sensor locations (matrix $\boldsymbol{C}$) are required. For model simulations, however, the $n \times m$ input matrix $\boldsymbol{B}$ would also be required and would contain inputs such as height differences between junctions, minor losses (valves, pumps), storage junctions (tanks), and the set values of flow or pressure at water sources and sinks (demand or reservoir junctions), and thus, at the boundaries of the system. Thus, for the intended goal of optimal sensor placement, based on state-space methodology, matrices $\boldsymbol{B}$ and $\boldsymbol{D}$ do not need to be specified and no temporal discretization of the system, Equations (7) and (8), is required.

The output vector $y$ is dependent on the $p_i$ junctions with head sensors and the $p_{ij}$ conduits with flow sensors, resulting in a binary pseudo-diagonal output matrix $\boldsymbol{C}$ with $p = p_i + p_{ij}$, where those elements of $\boldsymbol{C}$ are 1 if a sensor is present at that junction or in that conduit. For any chosen sensor configuration and accompanying output vector $y$

and output matrix $C$, the network is observable if the $pn \times n$ observability matrix $\mathcal{O}$ has full rank:

$$\mathcal{O} = \begin{bmatrix} C \\ CA \\ CA^2 \\ \vdots \\ CA^{n-1} \end{bmatrix} \tag{9}$$

A system that is observable allows a full reconstruction of the states over time from given input-output data. However, for large networks, the observability matrix $\mathcal{O}$ may be ill-conditioned, which would lead to an observability analysis that does not lead to accurate conclusions [33]. If the eigenvalues of matrix $A$ all have negative real parts, the system is called (asymptotically) stable. In that case, for each sensor configuration, the $n \times n$ observability Gramian $W_{\mathcal{O}}$ of the network can be calculated, after solving the discrete Lyanupov equation $A^T W_{\mathcal{O}} + W_{\mathcal{O}} A = -C^T C$, and is given by:

$$W_{\mathcal{O}} = \int_0^{\infty} \left( e^{A^T \tau} C^T C e^{A \tau} \right) d\tau \tag{10}$$

If $W_{\mathcal{O}}$, a symmetric matrix and unique solution to the discrete Lyapunov equation, is positive definite, that is, has all eigenvalues larger than zero, then the system defined by $A$ and $C$ is observable.

For linear, time-invariant systems, such as given by Equations (7) and (8), the sensitivity of the output $y$ with respect to the initial state $x(0)$ is given by $Ce^{At}$ [33]. Therefore, the observability Gramian $W_{\mathcal{O}}$ from Equation (10) can be interpreted as a Fisher Information Matrix, and it can thus be understood as a measure of information content. Its inverse, apart from a scaling factor, represents the uncertainty in the estimates of the states [34]. In the following, a norm of the observability Gramian $W_{\mathcal{O}}$ will be used as a measure for network observability. In this study, the smallest eigenvalue of $W_{\mathcal{O}}$ is chosen as norm, instead of a summarizing functional based on "optimality criteria" from optimal experiment design [35,36]. Georges showed that the eigenvalue-optimality criterion can be used to determine which system configuration, defined by $y$, as a result of the choice of matrix $C$, maximizes the observability [26]. This is achieved by quantifying the information content or "output energy" $\mathcal{E}(y)$ associated with each different sensor configuration, based on the real-valued non-negative eigenvalues $\lambda_{W_{\mathcal{O}}}$ of the corresponding observability Gramian $W_{\mathcal{O}}$:

$$\mathcal{E}(y) = \min_{k=1,\dots,n} \lambda_{W_{\mathcal{O}},k} \tag{11}$$

The smallest eigenvalue $\lambda_{W_{\mathcal{O}}}$ corresponds to a combination of states that are least observable. Choosing a sensor configuration that maximizes this minimum eigenvalue ensures maximum observability of this combination of least observable network states, thereby realizing the most meaningful increase in network observability. The sensor configuration that maximizes the output energy $\mathcal{E}(y)$ is the optimal sensor configuration $y_{opt}$ that maximizes the network's observability:

$$y_{opt} = \underset{y}{\operatorname{argmax}}(\mathcal{E}(y)) \tag{12}$$

Although the exact magnitude of the observability index, in this case the smallest eigenvalue, of a specific sensor configuration $y$ might not be preserved after our approximations, eigenvalue-optimality still allows for comparison of the observability index of different sensor configurations.

## 3. Results and Discussion

In order to illustrate the power of the state-space representation of hydraulic conduit networks introduced in Section 2, we will apply the proposed method, for optimal sensor

placement, to two hydraulic models of water distribution networks. Since the aim is to show the essence of the method, we restrict ourselves to small networks. In what follows, we assume a constant relative flow gradient $\varepsilon = 10^{-3}$ m$^{-1}$. See Appendix B for an analysis of the effect of the value of $\varepsilon$ on output energy and optimal sensor placement.

### 3.1. Example 1: Triangular Network

The small triangular network we study here is sketched in Figure 1a, and its properties are specified in Table 1. The network consists of three junctions $i = 1, 2, 3$ that are connected in a loop via conduits $ij = 12, 23, 13$ (Table 2). An additional conduit $ij = 41$ connects a reservoir $i = 4$ with constant head $H_4^0 = 243.84$m to node $j = 1$. The outgoing reservoir flow is assumed to be known, either inferred from measured reservoir volume or directly measured with a flow sensor on conduit $ij = 41$ The eigenvalue decomposition of the state matrix $A$ of the triangular network is detailed in Appendix C, showing that the system is asymptotically stable. The question, however, is: where could one extra sensor be best positioned?

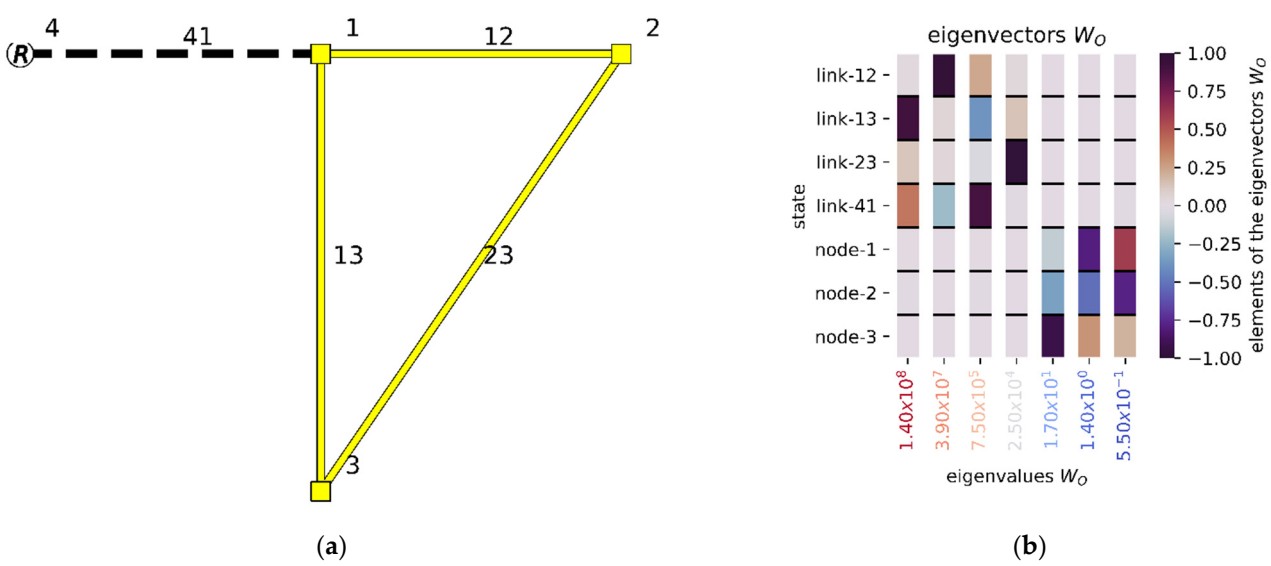

**(a)**　　　　　　　　　　　　　　　　　　　　　　　　　　　　　**(b)**

**Figure 1.** (**a**): schematic overview of the triangular network, where $R$ is the reservoir and conduit 41 has a flow sensor. (**b**): Eigenvalue decomposition of the corresponding observability Gramian $W_{\mathcal{O}}$ of the triangular network, with only a flow sensor in conduit 41, where each column represents an eigenvector of $W_{\mathcal{O}}$ and each column label at the bottom has the corresponding eigenvalue $\lambda_{W_{\mathcal{O}}}$.

**Table 1.** Network conduit properties of the triangular network.

| Conduit | Length [m] | Diameter [m] | Roughness | Flow [m³/s] | $\mathcal{X}_{ij}$[1/m²/s] | $\mathcal{Y}_{ij}$[m²/s²] | $\mathcal{Z}_{ij}$[s⁻¹] |
|---|---|---|---|---|---|---|---|
| 12 | 1524 | 0.2032 | 120 | $2.50 \times 10^{-2}$ | $4.53 \times 10^3$ | $2.09 \times 10^{-4}$ | $-4.85 \times 10^{-2}$ |
| 13 | 914.4 | 0.1524 | 80 | $1.10 \times 10^{-2}$ | $8.05 \times 10^3$ | $1.96 \times 10^{-4}$ | $-1.10 \times 10^{-1}$ |
| 23 | 243.8 | 0.3048 | 200 | $-1.48 \times 10^{-3}$ | $2.01 \times 10^3$ | $2.93 \times 10^{-3}$ | $-5.29 \times 10_{-4}$ |
| 41 | 304.8 | 0.3048 | 100 | $4.86 \times 10^{-2}$ | $2.01 \times 10^3$ | $2.35 \times 10^{-3}$ | $-3.74 \times 10^{-2}$ |

Since the triangular network is small, the corresponding observability matrix is well-conditioned. Eigenvalue decomposition of the observability Gramian $W_{\mathcal{O}}$ reveals that the three smallest eigenvalues are significantly smaller than the others (Figure 1a). Especially regarding the two smallest eigenvalues, the corresponding weights of the states of node 2 and node 3 are significantly higher than the weights of the other states in the eigenvectors associated with these two smallest eigenvalues. This indicates that the heads in nodes 2 and 3 are significantly less observable compared to the other states, and placing a sensor in either of these nodes will greatly improve the observability of the least observable part of

the network. Singular value decomposition of the observability matrix will give the same result, but $W_{\mathcal{O}}$ is less prone to ill-conditioning for large networks and thus presents a more robust indicator for optimal sensor placement.

**Table 2.** State matrix A of the triangular network including network topology.

| State | | Conduit | | | | Junction | | |
|---|---|---|---|---|---|---|---|---|
| | State | 12 | 13 | 23 | 41 | 1 | 2 | 3 |
| Conduit | 12 | $-4.85 \times 10^{-2}$ | 0 | 0 | 0 | $2.09 \times 10^{-4}$ | $-2.09 \times 10^{-4}$ | 0 |
| | 13 | 0 | $-1.00 \times 10^{-1}$ | 0 | 0 | $1.96 \times 10^{-4}$ | 0 | $-1.96 \times 10^{-4}$ |
| | 23 | 0 | 0 | $-5.29 \times 10^{-4}$ | 0 | 0 | $2.93 \times 10^{-3}$ | $-2.93 \times 10^{-3}$ |
| | 41 | 0 | 0 | 0 | $-3.74 \times 10^{-2}$ | $-2.35 \times 10^{-3}$ | 0 | 0 |
| Junction | 1 | $-4.53 \times 10^{3}$ | $-8.05 \times 10^{3}$ | 0 | $2.01 \times 10^{3}$ | 0 | 0 | 0 |
| | 2 | $4.53 \times 10^{3}$ | 0 | $-2.01 \times 10^{3}$ | 0 | 0 | 0 | 0 |
| | 3 | 0 | $8.05 \times 10^{3}$ | $2.01 \times 10^{3}$ | 0 | 0 | 0 | 0 |

For observability-based sensor placement, six options, and thus six different $C$ matrix, were considered: a head sensor in one of the nodes or a flow sensor in one of the conduits other than conduit 41, since the flow in conduit 41 is already metered. Notice from Equation (10) that the observability Gramian $W_{\mathcal{O}}$ is defined in terms of an inner product. In order to best visualize the output energy differences between various sensor placements, a square root color scale was used to put emphasis on the comparison between the output energies (Equation (11)) of the different sensor placements (Figure 2). As can be seen from Figure 2, sensor placement in junction 2 maximizes the output energy (smallest eigenvalue), closely followed by junction 3. As discussed above, this is in line with expectations, since nodes 2 and 3 were the states responsible for the smallest eigenvalues of the observability Gramian.

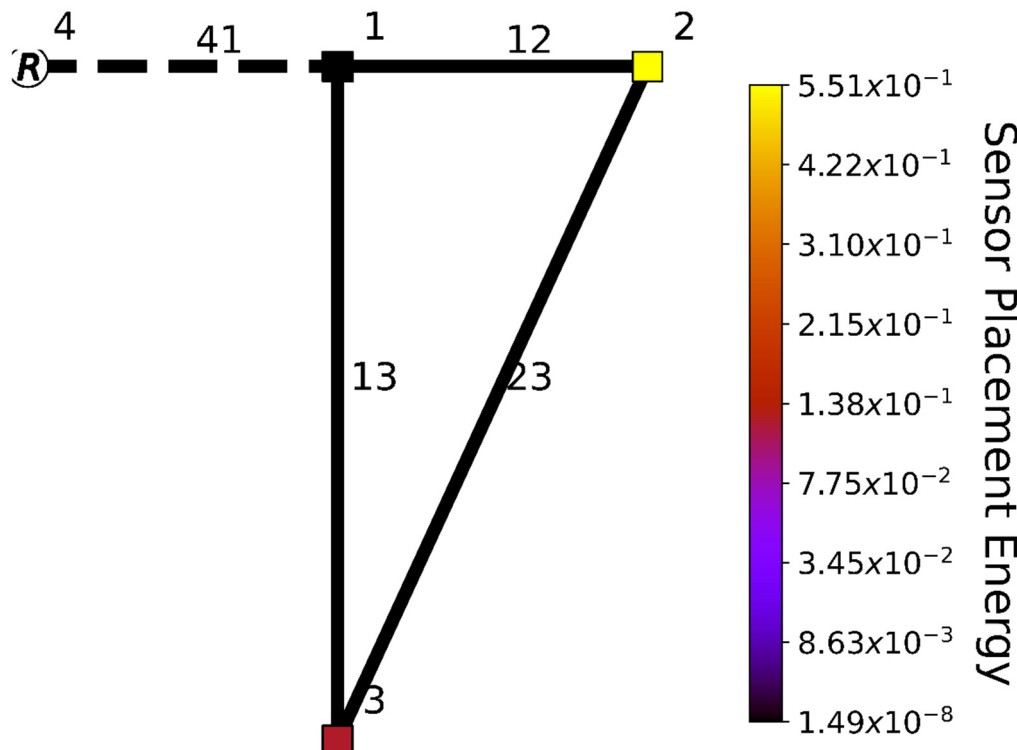

**Figure 2.** Triangular network, where $R$ represents a Reservoir, squares are junctions, and lines are conduits. Black dashed lines indicate a conduit with a flow sensor. Each additional possible sensor junction and conduit is colored based on the square root of the output energy (Equation (11)), corresponding with sensor placement in that specific junction or conduit.

### 3.2. Example 2: Net1 Case Study

In order to perform observability-based sensor placement, based on linear state-space models, estimates of the flow $\overline{Q}_{ij}$ through the network are required, using steady state hydraulic simulation of the system. However, these flows can differ significantly between scenarios. Therefore, an additional investigation was performed to determine the effect of hydraulic scenarios on resulting optimal sensor location. Optimal placement of one additional sensor for the EPANET chlorine decay model named 'Net1' was also considered (Figure 3a) [27]. Net1 is a network with one reservoir, tank, and pump, where the flow from/to the reservoir and the tank are assumed measurable, either directly or indirectly, from monitoring the reservoir and tank volumes. Depending on the time of day and the tank water volume, reservoir 9 is decoupled from the network, and tank 2 will act as a water source instead of a sink (Figure 3b). Scenarios with and without the reservoir will result in different optimal sensor locations. Therefore, placement of one additional sensor in Net1 was investigated with and without reservoir, at 08:00 and 20:00, respectively. Analysis at intermediate time instants, and thus, for different supplies and demands with corresponding steady state values of $H_i$ and $Q_{ij}$, did show different values of the output energy. However, this did not lead to changes in the optimal sensor location.

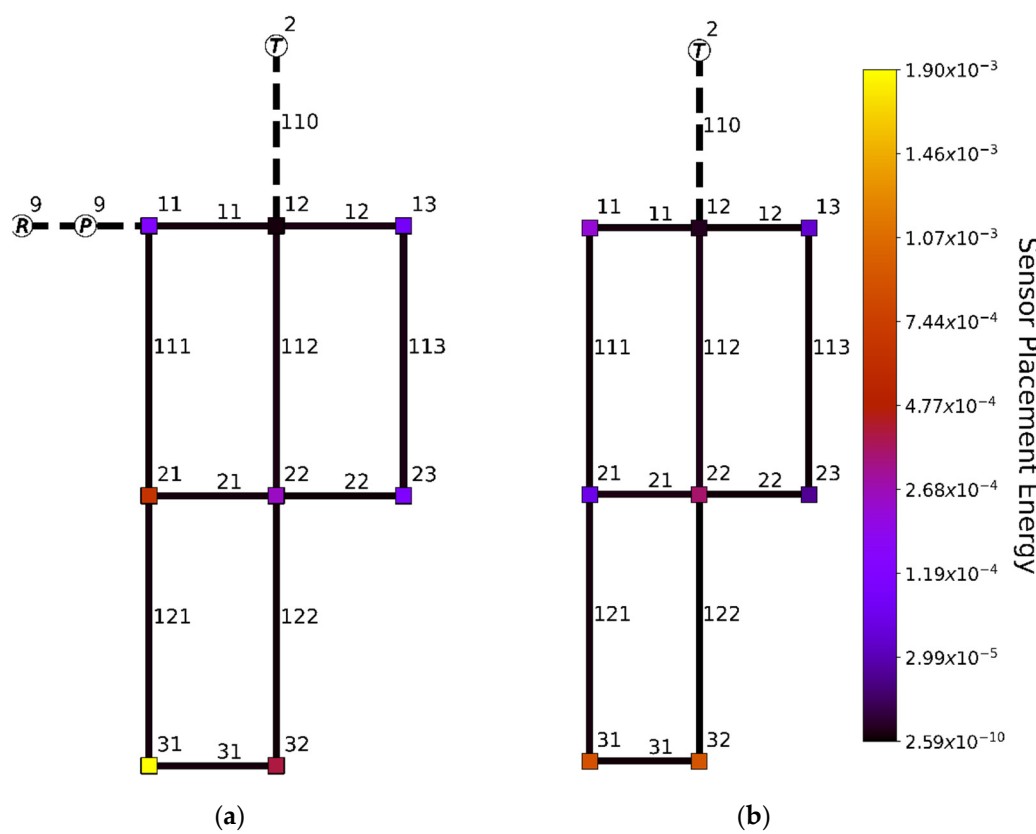

(**a**)  (**b**)

**Figure 3.** Optimal sensor placement for Net1 network with one tank (*T*), Reservoir (*R*), and pump (*P*) at 08:00 (**a**) and 20:00 (**b**). Black dashed lines indicate a conduit with a flow sensor. Each additional possible sensor junction and conduit is colored based on the square root of the energy corresponding with sensor placement in that specific state. At 20:00, conduit 9 is closed, thus decoupling reservoir 9 from the network.

Placement of one sensor was considered, in addition to the existing flow sensors, at conduit 110 and 9 (if the valve on conduit 9 is open). We found, for the case at 08:00, that a head sensor in junction 31 is optimal regarding network observability, as seen from the maximum output energy of this sensor configuration compared to alternative sensor placements (Figure 3). Depending on the valve configuration in the network, junction 32 could also be considered for sensor placement (Figure 3b). However, junction 31 is found

to be optimal for both network configurations, whereas junction 32 is not significantly more suited for sensor placement compared to 31, and it is significantly less suitable for the network configuration where the reservoir is connected to the network (Figure 3a). In both cases, the optimal sensor location is at the south end (bottom) of Net1. This is to be expected, since the original Net1 network only contains sensors in the north (top) of the network, so an additional sensor in the south enables network-wide insight. The fact that our placement procedure leads to optimal positions that are very close to each other, although the flow conditions are rather different, indicates that the linearization step does not significantly impact sensor placement performance. Since the valve configuration of the network for other time instances is similar to the configurations at 08:00 or 20:00, with only slight differences in network pressures and flows, optimal sensor placement for other time instances yields the same optimal sensor placement results.

If pressure changes are assumed instantaneous, and thus $\frac{dH_i}{dt} = 0$, only Equation (4) would remain for analysis of optimal sensor placement. Consequently, only flow sensor placement will be regarded optimal using this approach. In this case, the best choice is to place a flow sensor in the conduit with the largest resistance. Since both pressure and flow dynamics of the hydraulic system are included in the state-space model (Equation (7)), factors such as pressure wave velocity will effect sensor placement and thus result in more robust placement and investigation of pressure sensor placement in addition to flow sensor placement. In a practical sense, for full real-time reconstruction of all states, thus including the effect of water hammer, high speed (milliseconds—seconds) sampling sensors are needed, which are not commonly used. However, high speed sampling (in the order of milliseconds to seconds) is not considered a challenge nowadays, and the results presented vote for this strategy.

*3.3. Hanoi Network*

The triangular network was used to illustrate the state-space methodology, and the Net1 example network shows the robustness of state-space sensor placement with regards to changes in hydraulic scenario used for linearization. However, both networks are small theoretical networks. In order to investigate optimal sensor placement in a real network, the Hanoi network was used as a third case study. The Hanoi (Vietnam) network, is a drinking water distribution network with 34 conduits and 31 demand junctions, and it is used as a benchmark for optimal network design application [37,38].

Placement of one additional pressure sensor in the Hanoi network was successfully performed using the state-space methodology (Figure 4). When assuming the reservoir outflow conduit already contains a flow sensor, placement of a single pressure sensor in junction 25 is deemed optimal for increasing the observability of the network's least observable regions. A sensor on the border of the first and second network loop is, therefore, deemed optimal. Since the boundary of second and third loop is already metered via the flow sensor at the network reservoir, this configuration allows for metering all three loops as thoroughly as possible when placing just a single pressure sensor. This, in turn, will result in greatly improving the observability of each region of the network. In practice, this means that placement of one additional sensor will greatly benefit reconstruction of all network states (flows and pressures) and, therefore, will greatly supplement all network-wide methods and models, such as leakage detection algorithms or digital twins based on hydraulic models.

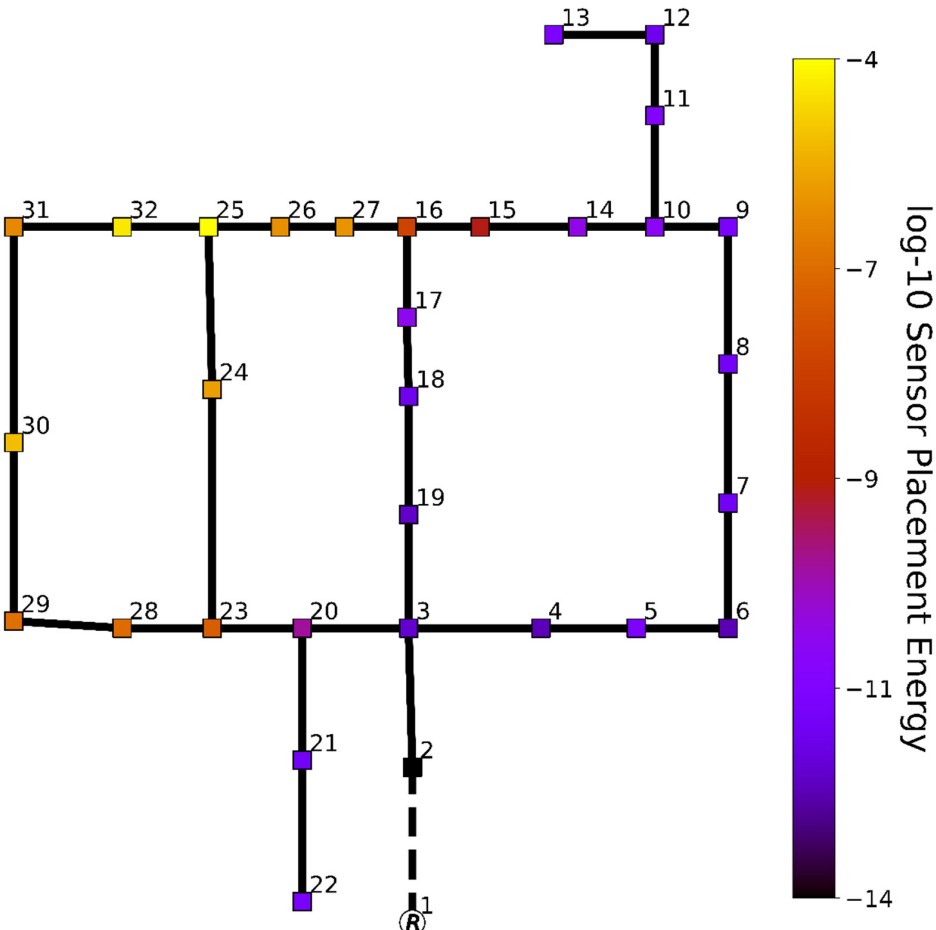

**Figure 4.** Comparison of possible pressure sensor placements in the Hanoi network. Possible pressure sensor junctions (squares) are colored based on 10-log output energy associated with placement of a sensor in that junction. Those conduits (black lines) attached to a reservoir (*R*) are already metered (dashed black lines).

## 4. Conclusions

Optimal sensor placement is a well-studied topic within the literature on smart water grids. However, the focus of these studies often lies on reducing placement uncertainty, as well as more computationally efficient optimization of the involved calculations. The method presented in this study expands the often used burst detectability-centric sensor placement criterion to an observability-based criterion. Our approach ensures that placement of additional sensors will provide more information about the entire network and will help improve hydraulic models or digital twins of the water distribution process. Using a state-space approach that takes flow, as well as pressure dynamics, into account, and does not rely on dynamic simulations, optimal sensor placement can be performed with limited computational efforts. Results based on three case studies indicate a robust sensor placement performance solely based on network observability. Additionally, the effect of piece-wise linearization of the system, as a result of changing hydraulic scenarios, is shown to not significantly impact sensor placement.

**Author Contributions:** C.V.C.G.: Conceptualization; Data curation; Formal analysis; Investigation; Methodology; Resources; Software; Supervision; Validation; Visualization; Roles/Writing—original draft. D.R.Y.: Funding acquisition; Project administration; Supervision; Writing—review & editin. J.M.: Project administration; Writing—review & editing. K.J.K.: Conceptualization; Funding acquisition; Methodology; Project administration; Supervision; Roles/Writing—original draft; Writing—review & editing. All authors have read and agreed to the published version of the manuscript.

**Funding:** This research received no external funding.

**Institutional Review Board Statement:** Not applicable.

**Informed Consent Statement:** Not applicable.

**Data Availability Statement:** EPANET model of the Hanoi (Bi et al., 2015; Fujiwara and Khang, 1990) is available via original publication. The triangular model data presented in this study is fully described in Table 1. The Net1 chlorine decay model is aviable from EPANET [27]. Custom Python 3.7 code was developed for this study.

**Acknowledgments:** This work was performed in the cooperation framework of Wetsus, European Centre of Excellence for Sustainable Water Technology (www.wetsus.nl, accessed on 7 October 2021). Wetsus is co-funded by the Dutch Ministry of Economic Affairs and Ministry of Infrastructure and Environment, the Province of Fryslân, and the Northern Netherlands Provinces. The authors would like to thank the participants of the research theme "Smart Water Grids" for the fruitful discussions and financial support.

**Conflicts of Interest:** The authors declare no conflict of interest.

**Notation:**

The following symbols are used in this paper:

| | | |
|---|---|---|
| Pressure | $p$ | Pa |
| Flow velocity | $V$ | $\mathrm{ms}^{-1}$ |
| Mass density of transported fluid | $\rho$ | $\mathrm{kgm}^{-3}$ |
| Elastic wave velocity | $c$ | $\mathrm{ms}^{-1}$ |
| Gravitational acceleration | $g$ | $\mathrm{ms}^{-2}$ |
| Angle of conduit versus horizontal | $\theta$ | rad |
| Darcy–Weisbach friction factor | $f$ | |
| Inside diameter of conduit | $D$ | m |
| Elevation | $z$ | m |
| Cross-sectional area of conduit | $A$ | $\mathrm{m}^2$ |
| Piezometric head | $H$ | m |
| Volumetric flowrate | $Q$ | $\mathrm{m}^3\mathrm{s}^{-1}$ |
| Hazen–Williams roughness coefficient | $C$ | |
| Relative flow gradient | $\varepsilon$ | $\mathrm{m}^{-1}$ |
| Observability matrix $pn \times n$ | $\mathcal{O}$ | |
| observability Gramian $n \times n$ | $W_{\mathcal{O}}$ | |
| Eigenvalue optimality output energy | $\mathcal{E}$ | |
| Eigenvalues of the observability Gramian $n \times 1$ | $\lambda_{W_{\mathcal{O}}}$ | |
| State vector $n \times 1$ | $x$ | |
| Output vector $p \times 1$ | $y$ | |
| Output vector associated with the optimal sensor configuration $p \times 1$ | $y_{opt}$ | |
| Input vector | $u$ | |
| State matrix $n \times n$ | $\boldsymbol{A}$ | |
| Input matrix | $\boldsymbol{B}$ | |
| Output matrix $p \times n$ | $\boldsymbol{C}$ | |
| Feedthrough matrix | $\boldsymbol{D}$ | |
| Network junction | $i$ | |
| Network conduit connecting junction $i$ and $j$ | $ij$ | |
| Total number of states | $n$ | |
| Number of junction head states | $n_i$ | |
| Number of conduit flow states | $n_{ij}$ | |
| Total number of states whose corresponding asset contains a sensor | $p$ | |

## Appendix A

Let us analyze the approximation of flow differences in a pipe, $\frac{Q_{ij(j)} - Q_{ij(i)}}{L} = \varepsilon Q_{ij(i)}$ with $\varepsilon \in \mathbb{R}$, in some more detail. This approximation is derived as follows. Assume the flow at the end points of a pipe is given by: $Q_{ij(j)} = (1 + \varepsilon L) Q_{ij(i)}$. Then,

$$\frac{Q_{ij(j)} - Q_{ij(i)}}{L} = \frac{(1 + \varepsilon L) Q_{ij(i)} - Q_{ij(i)}}{L} = \varepsilon Q_{ij(i)} \tag{A1}$$

With $\varepsilon$ the relative flow change per meter. Let the unsteady, nonuniformflow of a slightly compressible fluid in slightly elastic conduits, after linearization of the friction term around $\overline{Q}$, be described by the hyperbolic partial differential equation:

$$\frac{\partial}{\partial t} \begin{pmatrix} H \\ Q \end{pmatrix} + \begin{bmatrix} 0 & \overline{\mathcal{X}} \\ \overline{\mathcal{Y}} & 0 \end{bmatrix} \frac{\partial}{\partial x} \begin{pmatrix} H \\ Q \end{pmatrix} = \begin{pmatrix} 0 \\ -\overline{\mathcal{Z}} Q \end{pmatrix} \tag{A2}$$

where resistance $\overline{\mathcal{X}} = \frac{4}{\pi} \frac{c^2}{g D^2}$, conductance $\overline{\mathcal{Y}} = \frac{\pi}{4} g D^2$ and friction loss $\overline{\mathcal{Z}} = \frac{\pi}{4} \frac{10.67 g |\overline{Q}|^{0.852}}{C^{1.852} D^{2.8704}}$ are constants. After spatial discretization and defining the boundary conditions: $H(0,t) := H_0$ and $Q(L,t) := Q_1$

$$\frac{d}{dt} \begin{pmatrix} H(L,t) \\ Q(0,t) \end{pmatrix} + \begin{bmatrix} 0 & \overline{\mathcal{X}} \\ \overline{\mathcal{Y}} & 0 \end{bmatrix} \begin{pmatrix} \frac{H(L,t) - H_0}{L} \\ \frac{Q_1 - Q(0,t)}{L} \end{pmatrix} = \begin{pmatrix} 0 \\ -\overline{\mathcal{Z}} Q(0,t) \end{pmatrix} \tag{A3}$$

For easy of notation, we define: $H(t) := H(L,t)$ and $Q(t) := Q(0,t)$. Then,

$$\frac{d}{dt} \begin{pmatrix} H(t) \\ Q(t) \end{pmatrix} = \begin{bmatrix} 0 & \frac{\overline{\mathcal{X}}}{L} \\ -\frac{\overline{\mathcal{Y}}}{L} & -\overline{\mathcal{Z}} \end{bmatrix} \begin{pmatrix} H \\ Q \end{pmatrix} + \begin{pmatrix} 0 & -\frac{\overline{\mathcal{X}}}{L} \\ \frac{\overline{\mathcal{Y}}}{L} & 0 \end{pmatrix} \begin{pmatrix} H_0 \\ Q_1 \end{pmatrix} \tag{A4}$$

Using the approximation: $\frac{Q_1 - Q(t)}{L} = \varepsilon Q(t)$, gives

$$\frac{d}{dt} \begin{pmatrix} H(t) \\ Q(t) \end{pmatrix} = \begin{bmatrix} 0 & \varepsilon \overline{\mathcal{X}} \\ -\frac{\overline{\mathcal{Y}}}{L} & -\overline{\mathcal{Z}} \end{bmatrix} \begin{pmatrix} H \\ Q \end{pmatrix} + \begin{pmatrix} 0 & 0 \\ \frac{\overline{\mathcal{Y}}}{L} & 0 \end{pmatrix} \begin{pmatrix} H_0 \\ Q_1 \end{pmatrix} \tag{A5}$$

Both (A4) and (A5) are a two-dimensional linear time invariant system of the form $\frac{d}{dt} x(t) = \mathbf{A} x(t) + \mathbf{B} u(t)$. The eigenvalues $\lambda_A$ of the system matrix $\mathbf{A}$ in (A6) are given by,

$$\lambda_A = -\frac{\overline{\mathcal{Z}}}{2} \pm \frac{1}{2} \sqrt{\frac{L \overline{\mathcal{Z}}^2 - 4\varepsilon \overline{\mathcal{X}} \overline{\mathcal{Y}}}{L}} \tag{A6}$$

Hence, choosing $\varepsilon = 1/L$ will give the eigenvalues or poles of system (A4). Consequently, for $\varepsilon \geq \frac{L \overline{\mathcal{Z}}^2}{4 \overline{\mathcal{X}} \overline{\mathcal{Y}}}, 4 \overline{\mathcal{X}} \overline{\mathcal{Y}} \mathcal{Z}^2 L^2$ and all variables positive, both systems are asymptotically stable and have the same time constant $\frac{1}{|Re(\lambda_A)|}$. The approximate system becomes unstable for $\varepsilon < 0$.

## Appendix B

As expected, placing a sensor in the triangular network, in addition to the sensor in conduit 41, will always result in a higher output energy, as an additional sensor will increase system observability, independent of assumed value of the flow gradient $\varepsilon$ (Figure A1). For values of $\varepsilon$ within the interval $[10^{-6}, 1]$, the sensor configuration with a pressure sensor at node 2 or 3 maximizes the output energy. Therefore, in the case studies we chose $\varepsilon = 10^{-3} \text{m}^{-1}$, a good estimate regarding the application of optimal sensor placement.

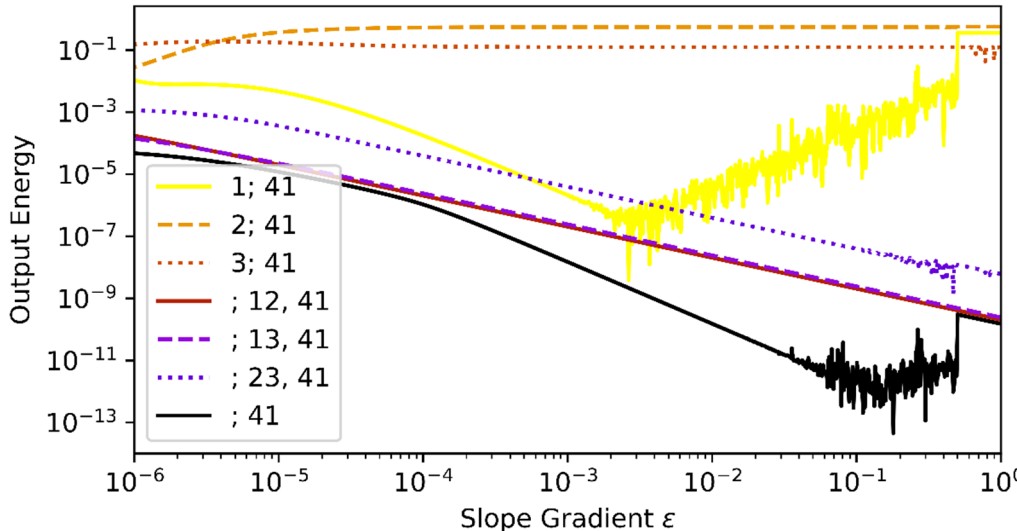

**Figure A1.** Effect of assumption of relative flow gradient $\varepsilon$ on the Output Energy.

### Appendix C

The eigenvalues $\lambda_A$ and eigenvectors $v_A$ (row-wise) of the state matrix $A$ of the triangular network are presented in Table 1. Notice from the eigenvalues that the system is asymptotically stable and will show oscillatory behavior, as expected. The "fastest" characteristic mode is related to the heads in the three nodes (see 5th row of eigenmatrix).

**Table 1.** Eigenvalues and accompanying eigenvectors of system matrix $A$.

| $\lambda_A$ | $v_A$ | | | | | | |
|---|---|---|---|---|---|---|---|
| | Link 12 | Link 13 | Link 23 | Link 41 | Node 1 | Node 2 | Node 3 |
| $-0.003 + 0.112j$ | $-0 + 0.001j$ | $0.001 - 0.001j$ | $0 - 0.037j$ | $-0.001 - 0.001j$ | $-0.018 + 0.046j$ | $0.709$ | $-0.702 - 0.032j$ |
| $-0.003 - 0.112j$ | $-0 - 0.001j$ | $0.001 + 0.001j$ | $0 + 0.037j$ | $-0.001 + 0.001j$ | $-0.018 - 0.046j$ | $0.709$ | $-0.702 + 0.032j$ |
| $-0.025 + 0.08j$ | $0.001 - 0.003j$ | $0.001 - 0.001j$ | $-0.002 - 0.002j$ | $-0.004 + 0.028j$ | $0.971$ | $-0.116 - 0.073j$ | $-0.191 - 0.029j$ |
| $-0.025 - 0.08j$ | $0.001 + 0.003j$ | $0.001 + 0.001j$ | $-0.002 + 0.002j$ | $-0.004 - 0.028j$ | $0.971$ | $-0.116 + 0.073j$ | $-0.191 + 0.029j$ |
| $-0.091$ | $0.004$ | $-0.014$ | $0.019$ | $-0.023$ | $-0.534$ | $0.23$ | $0.813$ |
| $-0.025 + 0.003j$ | $-0.005 + 0.001j$ | $-0.001$ | $-0.003 + 0.001j$ | $-0.019 + 0.002j$ | $0.102 + 0.012j$ | $0.716$ | $0.69 + 0.008j$ |
| $-0.025 - 0.003j$ | $-0.005 - 0.001j$ | $-0.001$ | $-0.003 - 0.001j$ | $-0.019 - 0.002j$ | $0.102 - 0.012j$ | $0.716$ | $0.69 - 0.008j$ |

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
