# Peer review of "Optimal Sensor Placement in Hydraulic Conduit Networks: A State-Space Approach"

_water, doi:10.3390/w13213105_

Round 1

Reviewer 1 Report

The authors present a theoretical work regarding the optimization of sensor placement in the hydraulic fluid transport network. In my opinion, the proposed framework is interesting and useful for practical use, which does not heavily rely on the expensive hydraulic simulations. From the last example demo in the manuscript, the applicability of this approach to solving real-world water distribution network issue was also verified. Based on these presented results, I suggest the acceptance of this manuscript for publication. In the below, I just have two minor questions for the authors: 

(1)  Page 3, Line 141, what is A'? is A' a typo?

(2) Page 4, Line 161, it seems the authors removed the convective terms in the governing equations (2). is that correct?  

Author Response

Thank you reviewer #1 for taking the time to review the manuscript. Below I hope to answer the two questions you pose:

(1)  Page 3, Line 141, what is A'? is A' a typo?

The mentioned apostrophe is a regular comma in the sentence and is not part of any equation. However, this can easily be mistaken for an apostrophe belonging to matrix A. Therefore, a space was inserted before the comma. The capital E used for Eq was also replaced with a small e. Original: “Furthermore, with [equation] and [equation], Eq (1) can be…”. Result: “Furthermore, with [equation] and [equation] , eq (1) can be…”.

(2) Page 4, Line 161, it seems the authors removed the convective terms in the governing equations (2). is that correct?

As stated on line 137, “Also, in many applications, the convective acceleration terms  and  are small compared to the other terms and may therefore be neglected (Chaudhry 2014).” The acceleration terms are used in eq (1) line 123, but are neglected in all following equations, due to their negligible impact.

Reviewer 2 Report

This study aims to propose a mathematical approach for optimizing sensor placement in hydraulic conduit networks on the basis of observability-based criterion, which is less common but may provide more effective information for decision making. The mathematical model ends up producing a convenient index, sensor placement energy, at junctions or parts of interest in given conduit network, allowing evaluation with consideration of fluid mechanics. Applications on two ideal scenarios and one realistic case are presented, showing the validity and effectiveness of the new approach. The manuscript is well written, and the introduction of the new approach is clear and in detail. The publication of this study in Water journal is recommended. A few comments should be taken into account:

1) Line 184: Please check if it is “to what extent”.

2) Figure 1: Does the color bar indicate λ? If yes, please add a label to it.

3) Figure 3: The color bar blocks the number 113. Please fix it.

Author Response

Thank you reviewer #2 for taking the time to review the manuscript. Below the three posed comments are addressed.

1) Line 184: Please check if it is “to what extent”.

It is indeed “to what extent” and not “to what extend”. I have corrected this mistake.

2) Figure 1: Does the color bar indicate λ? If yes, please add a label to it.

The figure vertically shows each eigenvector of the observability Gramian, where the colors correspond to the elements of these eigenvectors, which range from -1 to 1. The color bar to the right of the figure belongs to these vectors. The corresponding eigenvalues are shown below the eigenvectors, where each eigenvector is labelled with its corresponding eigenvalue. A label was also added to the color bar to clarify that the colors are the elements of the eigenvectors.

3) Figure 3: The color bar blocks the number 113. Please fix it.

The color bar has been moved slightly to the right, in order to prevent clipping with the labels.